# Locked in Structure: Sestrin and GATOR—A Billion-Year Marriage

**DOI:** 10.3390/cells13181587

**Published:** 2024-09-21

**Authors:** Alexander Haidurov, Andrei V. Budanov

**Affiliations:** School of Biochemistry and Immunology, Trinity Biomedical Sciences Institute, Trinity College Dublin, Pearse Street, D02 R590 Dublin, Ireland

**Keywords:** SESN2, GATOR2, mTORC1, conservation

## Abstract

Sestrins are a conserved family of stress-responsive proteins that play a crucial role in cellular metabolism, stress response, and ageing. Vertebrates have three Sestrin genes (*SESN1*, *SESN2*, and *SESN3*), while invertebrates encode only one. Initially identified as antioxidant proteins that regulate cell viability, Sestrins are now recognised as crucial inhibitors of the mechanistic target of rapamycin complex 1 kinase (mTORC1), a central regulator of anabolism, cell growth, and autophagy. Sestrins suppress mTORC1 through an inhibitory interaction with the GATOR2 protein complex, which, in concert with GATOR1, signals to inhibit the lysosomal docking of mTORC1. A leucine-binding pocket (LBP) is found in most vertebrate Sestrins, and when bound with leucine, Sestrins do not bind GATOR2, prompting mTORC1 activation. This review examines the evolutionary conservation of Sestrins and their functional motifs, focusing on their origins and development. We highlight that the most conserved regions of Sestrins are those involved in GATOR2 binding, and while analogues of Sestrins exist in prokaryotes, the unique feature of eukaryotic Sestrins is their structural presentation of GATOR2-binding motifs.

## 1. Introduction

Sestrins are a conserved protein family ubiquitously expressed across the metazoan kingdom. While vertebrates have three Sestrin genes (*SESN1*, *2*, and *3*), invertebrate genomes contain a single Sestrin gene (e.g., *cSesn* in *Caenorhabditis elegans*) [1]. In 1994, Buckbinder et al. discovered *SESN1* while screening for p53-responsive genes, naming it p53-activated gene 26 (*PA26*) [2]. In 2002, *SESN2*, linked to hypoxic stress response, was identified as hypoxia-inducible gene 95 (*Hi95*) [3]. Peeters et al. identified a homology between three genes and named them Sestrins as a tribute to the Italian city Sestri Levante, where the sequence analysis was conducted. Their 2003 paper mapped *SESN1*, *SESN2*, and *SESN3* to chromosomes 6, 1, and 11, respectively [4].

The biological significance of Sestrins is underscored by their role in protecting organisms against a variety of stresses [5], regulating cellular metabolism [6], modulating cell viability [7,8] and suppressing age-associated pathologies [5,9,10]. Sestrins achieve this by directly [11] and indirectly [12,13,14] supporting antioxidant defences and suppressing the mammalian target of rapamycin complex 1 (mTORC1) [6], which is a key signalling node for cell growth and metabolism. Sestrins’ localisation in the cytosol allows them to sense leucine and interact with the mTORC1-regulating machinery [15]. However, they also have been shown to localise to mitochondria [16] and nuclei [17] to support mitochondrial activity and antioxidant responses.

The activity of ubiquitously expressed Sestrins is controlled by leucine through binding with their leucine-binding pocket (LBP), which when bound, mitigates Sestrins’ ability to inhibit mTORC1 [18,19]. The level of Sestrin expression varies from tissue to tissue [9], and is thought to correspond to the anticipated leucine concentrations in different parts of an organism [19]. In many tissues under healthy growing conditions, Sestrin expression levels are relatively low and the available leucine relieves their inhibition of mTORC1. However, the upregulation of Sestrin genes in response to various stresses can override leucine sensitivity, through a marked elevation in Sestrin levels. DNA damage, oxidative stress, hypoxia, endoplasmic reticulum (ER) stress, energy deprivation, and amino acid starvation induce the expression of Sestrins by engaging several core stress-responsive transcription factors, including p53 [3,20], forkhead box O (FoxO) [21,22], nuclear factor erythroid 2–related factor 2 (Nrf2) [23], hypoxia-inducible factor 1-alpha (HIF1α) [3,24], CCAAT/enhancer binding protein (C/EBP) [25], and activating transcription factor 4 (ATF4) [26]. Responding to stress, Sestrin levels overwhelm its equilibrium with leucine, shifting the balance towards mTORC1 inhibition. Figure 1 illustrates the role of Sestrins in this pathway.

Recent structural studies have enhanced our understanding of Sestrin functions by clarifying their interaction mechanism with their key partner—the GATOR2 protein complex. This review examines the available structural data in combination with conservation analysis and discusses essential structural features of Sestrins that may be crucial to their function and evolution.

## 2. The Structure of Sestrins

Out of the three mammalian Sestrin genes, only the sole protein product of *SESN2* was structurally resolved [27,28,29]. In 2015, Kim et al. reported a structure of the SESN2 protein produced via a selenomethionine (SeMet) substitute approach [27]. SeMet was used to replace methionine in the human SESN2 protein to facilitate its structure determination via X-ray crystallography. Shortly after, in 2016, Sabatini et al. reported a detailed SESN2 structure stabilised by bound leucine [28]. Initially, a debate ensued over whether Sestrin is a leucine sensor [30,31]; however, it is suggested that the apo-structure of SESN2 has not yet been resolved [31]. This presents a challenge for recent advancements in artificial intelligence folding software, such as AlphaFold, potentially skewing their predictions. AlphaFold is trained on the available structures, and thus, all Sestrin structures predicted using this method are biased towards the leucine-bound SESN2 conformation, limiting the accuracy of molecular and binding modelling based on this approach.

The resolution of the SESN2 structure has established Sestrins as leucine-binding proteins [28], identified sites important for binding the SESN2 partner GATOR2 [27], and revealed that the two opposing Sestrin domains overlap, suggesting that Sestrins evolved from the fusion of a single protein.

Sestrins are multifaceted proteins. They are described to hold three domains: SESN-A (or N-terminal domain), SESN-B (linker domain), and SESN-C (or C-terminal domain), which is illustrated in Figure 2. SESN-A and SESN-C contain the functional sites of Sestrins, whereas the linker SESN-B is poorly conserved across Sestrin proteins. Although SESN-A and SESN-C each perform unique functions, they work together to bind and regulate GATOR2, and several studies have shown that they cannot accomplish this individually [15,28].

The Sestrin protein is almost entirely made of alpha helices. Notably, the regions most conserved among the Sestrin proteins fall within the α-helices. There are three highly conserved helical regions, separated by less conserved hinge sequences. Also, the N-terminal tails (NTT) of Sestrins are less conserved and exhibit a disordered structure [3].

We now know the regions that are important to Sestrin functions. In SESN-A, C125 is the site of Sestrin’s antioxidant activity. The S190 is important in GATOR2 binding. In SESN-C, the D406/D407 dyad was shown to be essential in binding GATOR2. SESN-C also houses several residues, importantly T377/386/426, that are essential in coordinating a leucine molecule (Figure 2). The next section investigates residue conservation across Sestrins to determine if these residues are conserved and whether their evolutionary preservation reflects their functional importance.

## 3. Sestrin Structure Conservation

To identify key residues within the Sestrin structure, we performed a multiple sequence alignment of 1006 Sestrin open reading frames, covering either a single Sestrin or all three Sestrins where available. We aimed to select a representative species from each genus of the metazoan kingdom. The alignment includes 575 genera, from nematode *Caenorhabditis* to *Homo*, encompassing 587 species. We avoided sequences listed as partial and selected the largest transcript where isoforms were available. Furthermore, using these sequences, we performed a Multiple Em for Motif Elicitation (MEME) motif discovery analysis [32], identifying conserved motifs six residues wide. To assess evolutionary significance, we repeated this analysis on non-metazoan Sestrin proteins, aligning 213 entries from 213 species across 131 genera. Our non-metazoan dataset includes unicellular eukaryotes like *Trypanosoma brucei* and *Naegleria gruberi*. The following section discusses domain conservation, with all positions and residue numbers referring to SESN2, unless stated otherwise. By comparing metazoan Sestrins to non-metazoan homologues, we aim to highlight the evolutionary conservation of motifs that are crucial to Sestrin function. The full analysis can be found in the Appendix A.

### 3.1. SESN-A: MAARQCSYL Motif

The site of Sestrin’s intrinsic antioxidant activity is C125 [11,27]. MAARQCYSL is the defining motif of the N-terminus in metazoan Sestrins. The MAARQ is the α-helix preceding the catalytic cysteine and CXYL is recognised by most domain databases as the anti-oxidative motif that is found in the SESN-A domain of Sestrins (Figure 3A).

The catalytic cysteine is 99.5% conserved across metazoan Sestrins. This conservation is further reinforced by a 90.1% retention in our non-metazoan analysis. Multiple mutation experiments of this residue display that it is essential for the redox activity of SESN2 [11,27].

It was shown that the residues surrounding C125 are highly hydrophobic, implying that the human SESN2 prefers hydrophobic substrates [27]. Furthermore, the authors proposed that Y127 and H132 are involved in the catalytic reduction process and proton relay of this site, similar to the alkyl hydroperoxidase D (AhpD) catalytic domain of *Mycobacterium tuberculosis.* The mutants of these sites also reduced the antioxidant activity of SESN2 [27]. In our analysis, Y127 and H132 are 93.9% and 82.2% conserved across metazoan Sestrins, respectively. While in non-metazoans Y127 is present in 91.5% of entries, H132 is present in only 1.9% of entries, suggesting that it was acquired later in metazoans.

### 3.2. SESN-C: Leucine-Binding Motifs

Leucine is an essential amino acid, acquired solely through diet as it cannot be synthesised endogenously. Precise leucine concentrations on the cellular level are difficult to estimate; however, a study on mouse livers investigated the effects of leucine starvation on a physiological level. In nutrient-rich conditions, it was shown that plasma leucine in mice ranged in concentrations of 250–300 µM. During full leucine starvation, this concentration dropped to 20 µM, at which mTORC1 was shown to be drastically inhibited. The study also showed that double SESN1 and 2-knockout mice were insensitive to leucine starvation, with their mTORC1 activity unaffected [33]. In a cell-free model, it was shown that the dissociation constant (Kd) for human SESN2 is 20 µM [18]. However, leucine concentration may vary depending on tissue context, and the relationship between leucine and Sestrin remains to be studied in greater detail in vivo.

During amino acid deprivation or fasting, intracellular leucine levels drop significantly, inducing a metabolic shift that involves Sestrin-mediated inhibition of mTORC1. This inhibition promotes autophagy, wherein cellular components, including proteins, are broken down into their constituent amino acids, such as leucine. Autophagy may temporarily replenish leucine levels, controlling Sestrin activity and cellular homeostasis until dietary intake is restored.

#### 3.2.1. TYNT Motif

The unique ability of SESN-C is to bind leucine. Overall, SESN-C is more strongly conserved than SESN-A, mostly due to the requirement of multiple residues to support leucine binding. For this binding, the first motif of note is TYNT. This motif is located on an extended hinge, which is said to act as a ‘lid’ to the leucine-binding pocket (LBP) (Figure 3B) [28]. 

Mutations of T374 and T386 abolish the binding of leucine [28]. In our analysis, the trinity of threonines is strongly conserved in metazoan Sestrins: T374 at 92.6%; T377 at 82.4%; and T386 at 92.4%. However, in non-metazoans, the MEME suite did not detect a motif at this position. Furthermore, these residues are less conserved in non-metazoans: T374 and T386 are conserved at 76.1% and 75.6%, respectively. T377 is present in only 4.2% of non-metazoan entries, and in 45.1% of cases, this site is a serine. This differential conservation of the lid residues suggests that the binding mechanism may have developed and been refined only in metazoan Sestrin proteins.

#### 3.2.2. HSEKVH Motif and Other LBP Residues

In addition to the ‘lid’ covering the LBP, the residues forming the base of the pocket are strongly conserved. The helix that forms the base of the LBP houses the HSEKVH motif, which is remarkably preserved (Figure 3C). Residues within this motif are some of the most conserved in metazoan Sestrins (90%+). E451 (HS**E**KVH), a crucial residue, is 99.5% conserved and has been shown to directly participate in leucine binding. The mutation of this residue abolishes leucine binding by SESN2 [28]. In non-metazoan Sestrins, this motif is less conserved; E451 shows only 80% conservation, while other residues in the motif are conserved at around 60%. Along with the absence of the TYNT motif in non-metazoans, this suggests that leucine-sensing motifs may have specifically evolved in metazoans.

Although not part of a defined motif, several residues have been identified as crucial for leucine binding in metazoan Sestrins. R390 is 92.4% conserved, and mutagenesis studies have shown it to be important in coordinating the carboxyl group of leucine [28]. Some distal sites from the LBP are also important. H86 of the SESN-A domain is 97.5% conserved, and was shown to be the residue that Y375 of the TYNT motif interacts with to seal the ‘lid’ over bound leucine [28]. L261, which is 95.9% conserved and located in the SESN-B linker domain, is the only highly conserved residue in SESN-B and is crucial for leucine binding, as shown by mutational analysis [18].

It was reported that SESN3 binds leucine very weakly compared to SESN1 and 2 [18]. It was also demonstrated that *Drosophila melanogaster* Sestrin (dSESN) binds leucine more weakly than human SESN2, with a dissociation constant of 100 µM compared to 20 µM for human SESN2 [18,34]. In our analysis, SESN3 or dSESN did not show any aberrations at the TYNT or HSEKVH motifs. Sestrins may have been less sensitive to leucine in earlier metazoans, and their sensitivity was optimised in response to the increasing complexity and metabolic needs of evolving metazoans.

### 3.3. SESN-A and SESN-C Co-Operate to Bind GATOR2

Although the two core domains of Sestrins are structurally similar, they exhibit key differences in the presentation of their conserved motifs. These distinctions are illustrated in Figure 4. While the motifs discussed in the previous section are unique to each domain and define the domain’s specific function, SESN-A and -C contain conserved motifs that co-operate to bind GATOR2. Two motifs are known to participate in this interaction, and they are among the most conserved motifs in both metazoan and non-metazoan Sestrins.

#### 3.3.1. WSLAEL: The SESN-A Site of Binding to GATOR2

The residue S190 has been proposed to participate in GATOR2 binding as the mutation of this residue abrogates the interaction [18]. The WSLAEL motif houses this residue and is found to be conserved among both metazoan and non-metazoan Sestrins (Figure 5A). The W189 (**W**SLAEL) residue is among the best conserved, with 98.9% conservation in metazoans and 98.6% in non-metazoans. Although S190 (W**S**LAEL) is conserved in 96.9% of metazoans, this site presents a serine in only 55.9% of non-metazoans, often featuring a threonine instead, in 32.4% of entries. Notably, there is also a remarkable conservation of E193, 97.6% in metazoans and 87.8% in non-metazoans. Only S190 has been determined to be important experimentally [18], and the other conserved sites of this motif remain to be studied further.

#### 3.3.2. DDYDY: The Electronegative Hinge of SESN-C

The SESN-A and SESN-C domains of Sestrins are structurally similar, but the unique feature of Sestrins is the presentation of a highly electronegative hinge in the SESN-C domain, in place of an α-helix found in the same position in SESN-A. The unique hinge presented at this position is home to the most conserved motif in Sestrins, the DDYDY motif. Across our metazoan analysis, this motif showed 100% retention of three of its residues and 99.7% conservation of its Y408 (Figure 5B). 

DDYDY is also remarkably conserved in non-metazoan Sestrins. The important D406 (**D**DYDY) is found in 53.5% of these Sestrins. Furthermore, in 25.4% of entries, glutamic acid replaces aspartic acid at this position, resulting in over 75% of non-metazoan Sestrins displaying an electronegative residue at this site. D407 (D**D**YDY) is 80.3% conserved in these Sestrins, with a further 9.9% presenting a glutamic acid, meaning that over 90% of non-metazoan Sestrins have an electronegative residue in this position (Figure 5B). The strong conservation of electronegative residues in the DDYDY motif across all eukaryotes highlights their critical role in Sestrin function.

Along with the electronegative residues, the hydrophobic tyrosines in this motif are also highly conserved. Y408 and Y410 are found in almost all metazoan Sestrins. Furthermore, in non-metazoans, Y408 is conserved in 86.4% of entries, while Y410 shows an even higher conservation at 96.2%. The high conservation of these tyrosines across all eukaryotes suggests that they play an important role. However, no experimental evidence has yet demonstrated the function of these tyrosines. They may stabilise the hydrophobicity of the motif during binding or could potentially play a role in regulating this site, possibly through phosphorylation by yet-to-be-determined kinases.

### 3.4. LAHRPW: The Conserved Motif at the Centre of Sestrin

The second most conserved motif after the DDYDY motif is LAHRPW, and there is no evidence that it serves a unique function. In metazoans, LAHRPW is remarkably conserved, displaying a 100% conservation in three of its residues, and over 98% conservation in two residues, L166 and A167. This remarkable conservation can be traced back to non-metazoan entries too. P170 and W171 are no less conserved in non-metazoans—over 95% for both of these residues (Figure 6A).

The analogous motif at this position on the SESN-C domain is not as widely conserved. IKTVAC is the motif in the SESN-C counterpart (Figure 6B, shown in blue) and while there are some highly conserved residues in this region (K422—99.1%; C430—98.3%; P432—99.9%; E433—91.8%), they do not form as tight a cluster of residues as the LAHRPW motif does. In non-metazoan entries, the conservation of the IKTVAC motif is unremarkable. Therefore, the stronger conservation of LAHRPW compared to its counterpart on the other core domain suggests its importance. It may serve as a key structural support as it is positioned centrally in the structure where the linker, SESN-C, and SESN-A domains meet. Alternatively, its conservation, comparable to the protein-interacting motif DDYDY, suggests it may be involved in an unknown interaction. In the available structures, the linker SESN-B is shown to cover access to this domain, and most of the LAHRPW residues are likely inaccessible, with only R169 and W171 being surface-exposed. Since only the leucine-bound SESN2 structure is resolved, our analysis may be biased, and the motif could be more surface-exposed in other Sestrin conformations.

## 4. The GATOR2–Sestrin Interaction

Sestrins are known to be potent mTORC1 inhibitors [6], primarily through their interaction with GATOR2. GATOR2 is a multiprotein complex composed of five subunits: WD Repeat Domain 59 (WDR59), Mios, WD Repeat Domain 24 (WDR24), SEH1 Like Nucleoporin (SEH1L), and Sec13. This large, 1.1 MDa, two-fold symmetric, octagonal complex presents eight pairs of WD40 β-propellers [35] and acts as a positive regulator of mTORC1 by sensing and integrating amino acid availability [36]. 

Active mTORC1 stimulates cell growth and proliferation [1,19]. Its activation involves two types of small GTPases, the RHEB and the Rag family. RagA/B and RagC/D heterodimers recruit mTORC1 to the lysosomal membrane, where RHEB binds and activates it [19]. GATOR1, a GTPase-activating protein for RagA/B, negatively regulates these GTPases and is inhibited by GATOR2. When the amino acid leucine is abundant, Sestrins dissociate from GATOR2, allowing GATOR2 to inhibit GATOR1, thus activating mTORC1. Under nutrient deprivation, Sestrins bind to GATOR2, inhibiting mTORC1 [37,38]. The structural features of GATOR2, especially its β-propeller domains, are essential for its interactions with amino acid sensors. Leucine availability disrupts Sestrins’ binding to GATOR2, relieving their inhibition of mTORC1 and leading to its activation [18].

We have previously shown that SESN2 interacts with GATOR2 proteins WDR24 and SEH1L but not with individual GATOR2 proteins or any other set of its components [37]. With the recent resolution of the GATOR2 complex, the authors have also shown that the β-propeller domains of WDR24 and SEH1L are sufficient for the SESN2 interaction. The authors generated a fused single chain of the WDR24 and SEH1L β-propellers, and this fusion of truncated proteins stably bound SESN2 [35]. This suggests that SESN2 makes multiple contacts with this arrangement, possibly with both SEH1L and WDR24, nestling into the unique interface presented when these two proteins are joined together. It is also worth noting that GATOR2 possesses two WDR24-SEH1L β-propeller regions, potentially enabling the binding of two Sestrin molecules simultaneously. The appearance and location of the WDR24-SEH1L β-propeller regions are visualised in Figure 7A.

We attempted a similar multiple sequence analysis of WDR24 and SEH1L as with Sestrin proteins (Appendix A). Figure 7B(a–c) visualises the most conserved residues in the WDR24-SEH1L β-propeller arrangement. The β-propellers of WDR24 and SEH1L integrate into the GATOR2 complex via blade donation, as shown in Figure 7B(c). The majority of the most conserved surface residues face towards the circular interface formed by the WDR24-SEH1L interaction. It is therefore plausible that this is the interface that Sestrins target for binding. Recent studies employing AF2-multimer, a specialised version of the AlphaFold2 (AF2) algorithm designed for modelling protein–protein interactions within complexes, have proposed that SESN2-GATOR2 binding occurs along this interface [39]. Despite the insights, it is worth considering that the available structural Sestrin data are biased towards the leucine-bound conformation, which may lead to inaccuracies when using this method of modelling. However, the authors of the same study have provided mutational analysis showing two additional residues that may be important in this interaction, L351 and D364, which our metazoan analysis shows as 96.2% and 97.7% conserved, respectively.

Mapping all residues with evidence of the GATOR2-SESN2 interaction, Figure 7C(a) displays a hypothetical orientation for this interaction, while Figure 7C(b) shows that these residues are located along the same face of SESN2. 

In line with this hypothesis, the WSLAEL motif containing S190 may bind along the conserved SEH1L face. To maintain interaction with WDR24, multiple points of contact are required, including the well-characterised D406/D407 of the DDYDY motif. Mutational analysis shows that the loss of any of these residues on SESN2 disrupts GATOR2 binding. Despite the multiple points of contact available to SESN2, the cumulative strength of all these interactions might represent the minimum required for stable binding to GATOR2.

## 5. The Ancestry of Sestrins

In earlier studies, Sestrin was identified as a distant homologue of the *Mycobacterium tuberculosis* AhpD [11], which participates in the reduction of the bacterial peroxiredoxin AhpC [40]. However, in eukaryotes, this antioxidative mechanism has evolved into a more complex and intricate network compared to the prokaryotic counterparts. Eukaryotic AhpC analogues are known as peroxiredoxins (Prx), and this widely distributed family of enzymes can be divided into 1-Cys and 2-Cys Prxs, based on the number of conserved cysteine residues that participate in their catalytic cycle [41]. The 2-Cys Prxs can be assisted by thioredoxins (Trx), which reduce their disulphide bond following redox scavenging [42]. In eukaryotes, peroxiredoxins are sensitive to overoxidation, and when reactive oxygen species (ROS) levels are high, peroxiredoxins become overoxidised and are reversibly inactivated [43]. Unlike their prokaryotic counterparts, which undergo irreversible inactivation under severe oxidative stress, eukaryotic peroxiredoxins developed the ability to be reactivated. Sulfiredoxins (Srx), which are unique to eukaryotes, regenerate these overoxidised Prxs [44].

Sestrins were first proposed as barriers to Prx overoxidation. While it was demonstrated that Sestrins do not directly protect Prxs from overoxidation, their presence was shown to substantially increase the rate of recovery of overoxidised Prxs [11]. However, later studies demonstrated that Sestrins are not directly involved in reducing overoxidised Prxs [45]. Furthermore, it has been shown that Sestrins preferentially use their catalytic cysteine to directly neutralise hydrophobic ROS, such as cumene hydroperoxide, an alkyl hydroperoxide with a bulky hydrophobic group, rather than directly reducing smaller ROS, such as hydrogen peroxide [27]. The evolutionary niche that Sestrins’ antioxidative function occupies is not yet fully understood. Their antioxidative properties may be crucial in specific contexts, such as protecting their binding partners against hydrophobic ROS or neutralising particular types of ROS. Furthermore, the antioxidative mechanism of Sestrins differs significantly from that of AhpD, highlighting a gap in our understanding of the evolutionary development of Sestrins.

The core domains of Sestrin, SESN-A and SESN-C, each structurally resemble an AhpD monomer, and thus it was hypothesised that Sestrins are descendants of AhpD-like proteins. On the basis of the resolved structure of AhpD from *Streptococcus pneumoniae* (spAhpD), we attempted to search for structures that resemble the spAhpD dimer in order to find bacterial precursors of Sestrins. We used the programme PYMOL, the protein folding model Foldseek [46], and its internet portal AlphaFold Clusters by Steinegger Lab [47] to search for structural analogues of Sestrins. We took the quaternary structure of the spAhpD dimer and fused it in silico using PYMOL. The resultant fused dimer of spAhpD demonstrated a remarkably similar structure to SESN2 with a Root Mean Square Deviation (RMSD) of 3.2. Using AlphaFold Clusters, we searched the database using the PDB file of the artificially fused dimer and found several clusters of bacterial proteins that resemble the Sestrin architecture. Figure 8 displays several of these structural analogues and the hypothetical evolutionary journey from monomer to Sestrin, from left to right.

The first step of the hypothetical evolution of Sestrin would be the fusion of two AhpD-like molecules. One of the structural analogues we found is the protein product of the gene *yciW* of *Escherichia coli.* The entry was found in AlphaFold Cluster: A0A3P6L468, a cluster of putative amidase/amidotransferases. It was shown that this protein may be involved in the metabolism of L-cysteine into L-homocysteine, and potentially L-methionine [48]. The protein YciW contains two AhpD-like domains. In the N-terminus, it possesses two cysteines, but they are five residues apart, unlike AhpD which has a short distance of two residues between its catalytic cysteines. In the C-terminus, there are two cysteines as well, but they are eight residues apart. This suggests that this protein could share a common ancestor with AhpD, yet it has adapted the AhpD architecture and catalytic cysteines for a different function. Carboxymuconolactone decarboxylase-like (CMD) protein from *Candidatus Rokubacteria* bacterium, a member of the uncharacterised and uncultivated phylum, has a primary sequence resembling an AhpD-like fused dimer. The alignment of the N-terminus to C-terminus of this protein shows that the two domains closely resemble each other, suggesting that it is an example of a recently fused AhpD-like dimer.

The differentiation of one of the AhpD-like domains of such a fused dimer could be the first step towards a multifaceted function, much like what Sestrins perform. An alkyl hydroperoxidase from *Corynebacterium ureicelerivorans* bears a striking resemblance to Sestrin, so much so that the AlphaFold Cluster: A0A7S7ZDV4 displayed non-metazoan Sestrins and SESN3 entries in the same group. This protein shows a striking feature: while one domain fully resembles AhpD, the other has evolved into an AhpD-like domain lacking catalytic cysteines. The C-terminal domain contains two cysteines, similar to AhpD, but the N-terminal domain, while retaining a helix-turn-helix architecture, lacks cysteines. Similarly, alkyl hydroperoxidase from *Comamonas aquatica* seems to have lost cysteines from one of its AhpD-like domains. While possessing two cysteines in its C-terminus, the N-terminus is an AhpD-like structure without cysteines. These proteins may be capable of performing the function of AhpD, but they carry a pseudo domain, which could be free for experimental evolution. 

Despite finding several proteins that closely resemble Sestrin, in that they have two AhpD-like domains that adopt the Sestrin structure, we could not find a striking structural feature of Sestrins that may define them: an electronegative hinge in place of the cysteine active site. In our search, it appears that Sestrin is the exclusive protein in eukaryotes that has preserved the AhpD-like fused dimer structure. The key difference between Sestrins and prokaryotic analogues is the deletion of an α-helix at the helix-turn-helix structure of the C-terminal domain, as well as the presence of an electronegative hinge in its place, which houses the conserved DDYDY motif. Therefore, the key defining feature of Sestrins is the adaptation of the AhpD structure to present this highly electronegative hinge.

**Figure 8 cells-13-01587-f008:**
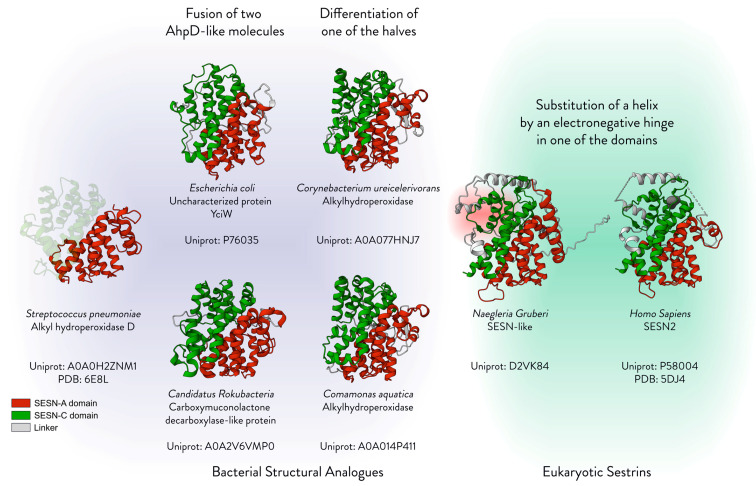
The ancestry of Sestrins. The figure displays the bacterial structural analogues of Sestrins. The hypothetical evolutionary route is illustrated from left to right, beginning with the monomeric spAhpD and ending with the human SESN2. The structures of the bacterial proteins were aligned to SESN2 using PYMOL, and Mol* Viewer was used to generate cartoon representations at the same angle for visual clarity. PDB IDs are annotated and for unresolved structures, the AlphaFold predictions were used [49]. The RMSD values of the structural alignment to SESN2 are as follows: spAhpD—RMSD 3.353093 over 152 residues; Coma_aqua—RMSD 2.967612 over 144 residues; Cory_urei—RMSD 3.812889 over 152 residues; Candid_Roku—RMSD 4.020528 over 144 residues; YciW—RMSD 6.050121 over 144 residues; SESN_Naeg—RMSD 3.472114 over 184 residues. Red—SESN-A; Green—SESN-C.

## 6. Discussion

The evolutionary time gap between prokaryotes and eukaryotes is vast, estimated to be between 1.5 and 2 billion years [50,51], making it challenging to identify Sestrin homologues in prokaryotes. While primary sequences of proteins can differ significantly, their structures are often conserved to a much greater extent [52,53]. The prokaryotic analogues we present in this review might not have a direct relation to Sestrins, but their occurrence can provide insights into Sestrin evolution. Sestrins likely share a common ancestor with AhpD, which may have gone through several iterations before evolving into the eukaryotic Sestrin as we know it. For instance, the YciW protein in *Escherichia coli*, which contains two AhpD-like domains with catalytic cysteines spaced further apart than in AhpD, suggests an adaptation from the original AhpD structure. Similarly, Sestrin progenitors, like the AhpD from *Corynebacterium ureicelerivoransa*, may have lost a cysteine in one domain. This loss could have allowed one domain to ‘experiment’ with new functions, leading to the Sestrin domain arrangement with two or more complementary functions.

The most conserved domains in Sestrins relate to GATOR2 binding. Sestrins that are found in early eukaryotes such as *Naegleria gruberi, Trypanosoma brucei,* and *Dictyostelium discoideum* contain motifs closely resembling DDYDY and WSLAEL. It is likely that early Sestrins were evolutionarily selected based on their relationship with GATOR2 or GATOR2-like proteins. The subunits of GATOR2 and its yeast homologues SEACAT have been suggested to resemble vesicle-coating scaffolds, implying an ancestral relationship that may have involved membrane-associated functions [54]. Furthermore, the WDR24-SEH1L β-propeller blade donation is analogous to the interaction of SEH1L with NUP85 and NUP145C in the nuclear pore complex [55,56]. Although this ancestral relationship has been noted, its significance for the evolution of GATOR2 and Sestrin requires further investigation. Early Sestrins may have influenced the function of GATOR2 or similar proteins, either by supporting or inhibiting their activity. Sestrins have been shown to be specific to decomposing hydrophobic ROS [27]. Given that their partners are ancestrally linked to membranes, Sestrins could have utilised their binding capabilities to co-localise with these proteins, thereby protecting essential membrane-associated machinery from oxidative damage, such as hydrophobic ROS, which can compromise membrane integrity. Understanding the evolutionary trajectory of GATOR2 and its components could provide insights into the origins and evolution of Sestrins.

Sestrins are widely prevalent across metazoans; however, Sestrins are missing in many non-metazoan organisms, including the model organism *Saccharomyces cerevisiae*. Despite containing homologues of GATOR2 and GATOR1—the SEACAT and SEACIT complexes—*Saccharomyces cerevisiae* notably lacks Sestrins [54,57]. Meanwhile, early Sestrin proteins with motifs resembling DDYDY, WSLAEL, and CSYL can be found in free-living amoeboids like *Naegleria gruberi* and in slightly more complex organisms such as *Yarrowia lipolytica*, a yeast species known for its robust metabolic capabilities [58]. The adoption of Sestrins may be tailored to the specific needs of an organism. *Saccharomyces cerevisiae*, unlike mammals, produces leucine endogenously [59] and may not need a metabolic modulator like Sestrin. Sestrins appear to be adopted by organisms that must deal with rapid environmental changes. For example, Sestrin is found in *Trypanosoma brucei*, which cycles through the environment of a host bloodstream and the intestinal gut of the tsetse fly. Similarly, Sestrin is present in organisms with multiple life cycles, such as *Dictyostelium discoideum* [60], where they are crucial for adaptation to starvation at specific life cycle stages. However, Sestrins are absent from the plant kingdom and the ‘higher fungus’ phylum *Basidiomycota*, commonly known as mushrooms. The common traits among early Sestrin adopters and metazoans are the constant environmental change and the need to rapidly adapt to stress. Therefore, Sestrins may have been adopted by organisms frequently facing rapid environmental change, becoming key components of their stress response mechanisms.

Recent research on Sestrins has focused heavily on their interaction with GATOR2, but the mechanism behind this interaction remains unclear. One of the major unanswered questions is the mechanism by which Sestrins inhibit the activity of GATOR2. How does Sestrin binding cause GATOR2 to change and bring about its inhibition? What is the stoichiometry of SESN-GATOR2 binding, and does the binding of one or two Sestrins to GATOR2 have different functional outcomes? What is the nature of the binding site, and can it be modified through post-translational modifications, thereby providing a rapid regulation of the SESN-GATOR2 interaction? Identifying the precise binding site of Sestrins on GATOR2 could provide crucial answers to these questions.

One of the most notable discoveries in recent years regarding Sestrins is their ability to bind leucine. However, many questions remain unanswered. How exactly does leucine binding prevent Sestrins from interacting with GATOR2? Does leucine binding trigger a major conformational shift? What does the apo-structure of Sestrin look like? Could leucine binding alter other functions or the localisation of Sestrins in addition to affecting GATOR2 binding?

Another unresolved question is the specific role of Sestrins in the antioxidant response. Although they can neutralise certain ROS and may support other antioxidative pathways, their exact physiological and evolutionary niche remains unclear. It is also uncertain how GATOR2 binding relates to Sestrin’s antioxidative activity, specifically whether these functions evolved separately or if their co-operation was instrumental to Sestrin’s evolution.

## 7. Conclusions

Sestrins are multifaceted proteins composed of two core domains that closely resemble each other but are specialised for distinct functions: SESN-A for antioxidative activity and SESN-C for leucine binding. Both domains work together to bind GATOR2, which dramatically impacts cellular processes by inhibiting mTORC1. Sestrins likely originated from an ancient fusion of two AhpD-like proteins, resulting in two AhpD-like domains within one protein. SESN-A retains a structural and functional similarity to the antioxidative AhpD, while in SESN-C, an α-helix near the active site was replaced by a hinge, thereby altering its function. DDYDY, the most conserved motif of Sestrins, is located on this hinge and is central to their function. Sestrins can be defined by the presence of this hinge structure and the conserved electronegative motif, such as DDYDY, at that position.

The most conserved motifs in Sestrins are involved in binding GATOR2 proteins, a function likely critical to their evolutionary development. Studying these conserved domains and the co-evolution of Sestrin partners may reveal unknown functions and help define the evolutionary niche of Sestrins. Sestrins’ ability to sense leucine likely developed after acquiring the GATOR2-binding function. While sequences resembling GATOR2-binding motifs can be elucidated in non-metazoan Sestrins, the leucine-sensitive sequences show lower conservation.

The conserved relationship between Sestrins and GATOR2 underscores a billion-year partnership that has shaped fundamental cellular processes through the regulation of mTORC1. Deciphering this relationship is crucial for understanding the origins of Sestrins and GATOR2 as well as their roles in cellular function and evolution.

## 8. Methods

### 8.1. Selection of Sequences for Alignment

Protein PSI-BLAST [61] from NBCI was used to search for sequences homologous to human SESN2 (NP_113647.1) and *Caenorhabditis elegans* SESN (CCD68266.1). In BLAST options, ‘Max target sequences’ was adjusted to 10,000. The taxonomic tab in the results of the search was used to select a species from each genus in the metazoan group. For non-metazoans, protein PSI-BLAST was performed similarly on the sequence of *Naegleria gruberi* SESN (XP_002675647.1), and the taxonomic tab was used to select sequences excluding metazoan. Partial and truncated sequences were excluded from analysis. In cases of multiple isoforms, the longest transcript was selected.

### 8.2. Alignment of Sequences and Analysis

The EMBL-EBI Clustal Omega [62] was used to align sequences and the programme Jalview [63] was used to inspect the alignment. Annotations such as consensus for each position were retrieved from Jalview using the export annotations option, and the results were exported to .csv file format. The data was processed using Microsoft Excel, until it was presented in a spreadsheet for easier viewing (as seen in Appendix A). Residues with low conservation were filtered out in Excel to maintain ease of viewing. For metazoan entries, residues with less than 40% conservation were ignored. For non-metazoan entries, residues with less than 25% conservation were ignored. The cell shading for consensus percentage was set using Excel conditional formatting option ‘Graded Colour Scale’, ‘3-Color Scale’ with Minimum & Maximum being the lowest and highest Values, and the midpoint being set to ‘Percentile’ value 70.

### 8.3. MEME Motif Discovery

The Multiple Em for Motif Elicitation (MEME) suite [32] was used for motif discovery and the same sequences that were used for alignment were supplied to the programme. The requested number of motifs was 15 and the motif Minimum & Maximum width was 6.

### 8.4. Generation of Fused AhpD Dimer for Foldseek

The PDB file 6E8L was loaded into PYMOL [64] and then chains A and B were combined into a single object. Residue numbers of chain B were altered n+181 residues, and chain B was altered to chain A. The last residue of one chain was bonded to the other using the ‘bond’ command. The resultant object was saved as a PDB file. The file was used on the AlphaClusters (AFDB) web portal to search for structure using the PDB file.

Mol* Viewer was used to view, colour, and generate the high-definition images of structures shown in the relevant figures [65].

## Figures and Tables

**Figure 1 cells-13-01587-f001:**
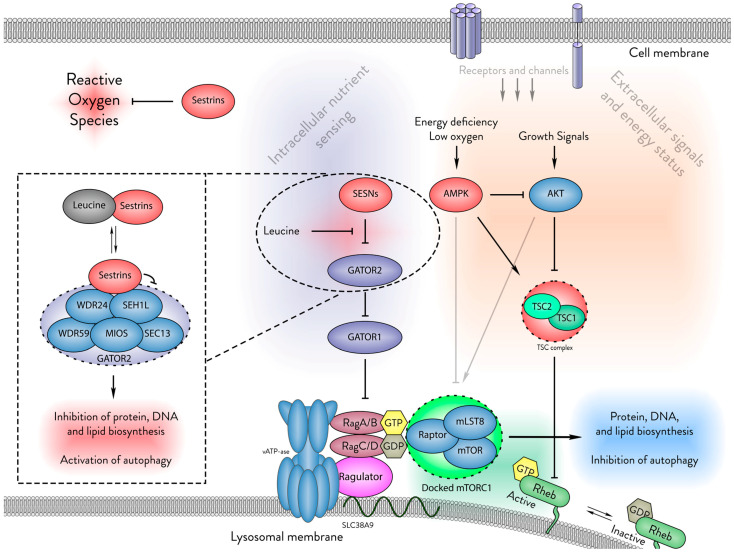
The cellular functions of Sestrins. Sestrins play a critical role in cellular homeostasis by neutralising reactive oxygen species and sensing intracellular leucine levels. In the absence of leucine, Sestrins bind to the GATOR2 complex, thus inhibiting mTORC1. This binding prevents mTORC1 from docking on the lysosomal membrane, thereby suppressing protein, DNA, and lipid biosynthesis while promoting autophagy. The dotted lines in this figure highlight the role of Sestrins in this intracellular nutrient-sensing pathway. This review focuses on the evolutionary conservation of Sestrins and the structural basis of their interactions within this pathway.

**Figure 2 cells-13-01587-f002:**
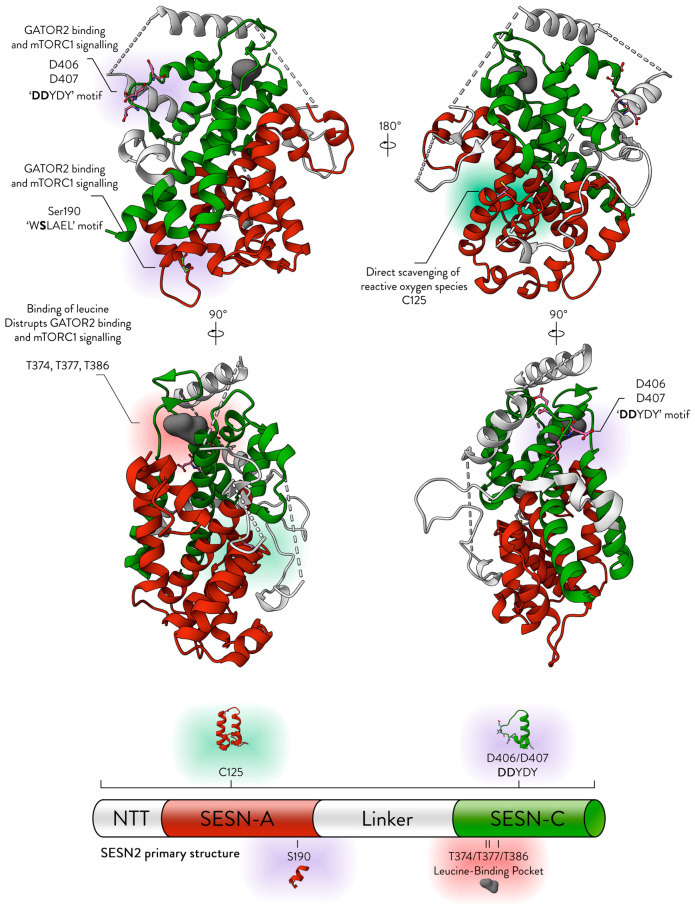
The structure of SESN2. The structure is displayed from four different angles (PDB ID: 5DJ4). A cartoon of the primary structure is displayed below. NTT—N-terminal-tail. The red, white, and green colours correspond to the domains they represent on the structure. Red—SESN-A; Green—SESN-C; White—linker. Important sites are annotated.

**Figure 3 cells-13-01587-f003:**
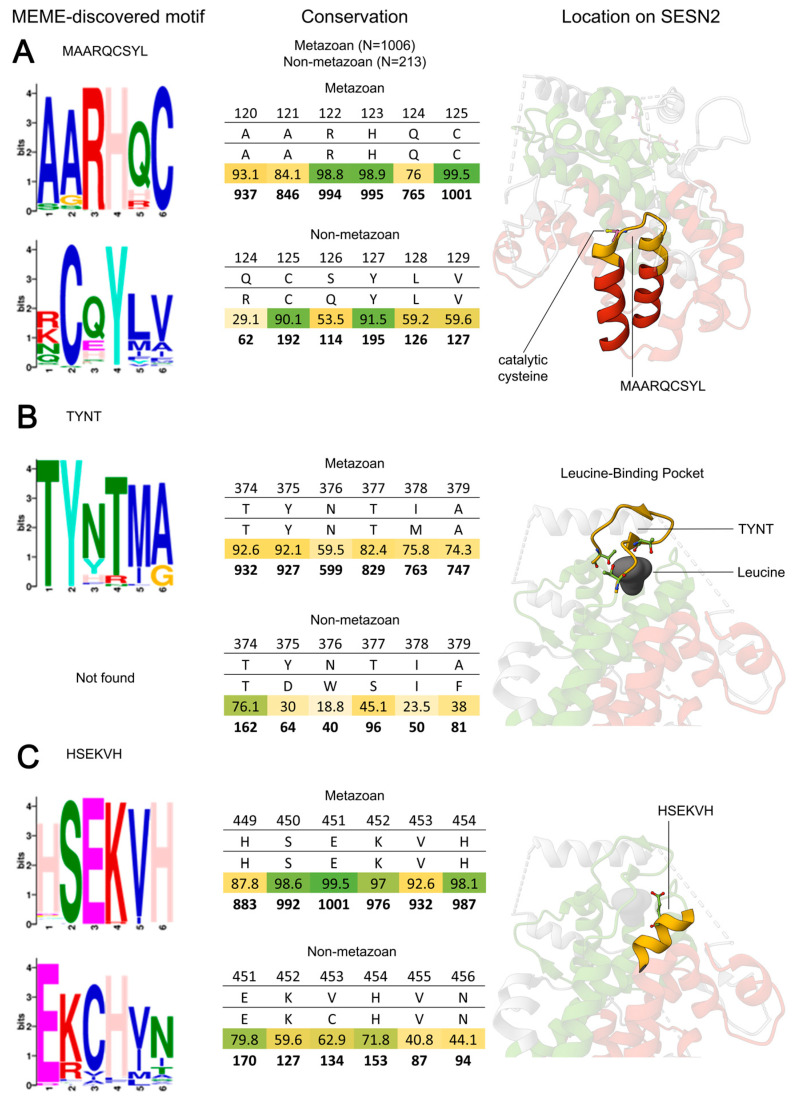
Conservation of Sestrin motifs responsible for the unique functions of SESN-A and SESN-C. The readout of the MEME suite motif discovery tool is displayed in a histogram of letters. An Excel representation of our alignment analysis is presented. The rows represent the following: row 1—position on SESN2; row 2—residue on SESN2; row 3—alignment consensus at this position; row 4—the % of entries showing consensus; row 5—the number of entries showing consensus (Metazoan N = 1006, Non-metazoan N = 213). The position on the SESN2 structure (PDB ID: 5DJ4) is shown on the right. (**A**) Motif MAARQCSYL that is responsible for the unique function of SESN-A. (**B**,**C**) Motifs that are implicated in the unique function of SESN-C.

**Figure 4 cells-13-01587-f004:**
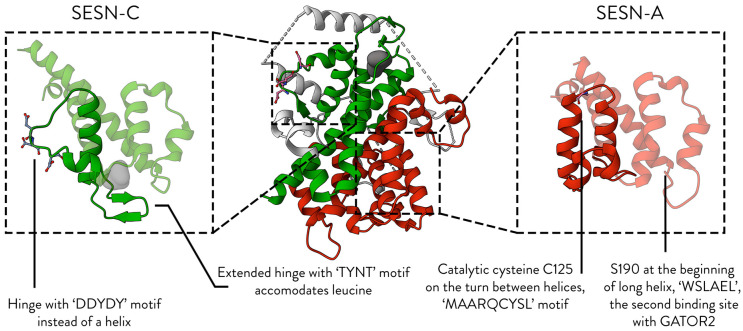
The key structural difference between SESN-A and SESN-C domains. This figure highlights the key structural differences between the two core domains of SESN2 (PDB ID: 5DJ4). Key differences are annotated.

**Figure 5 cells-13-01587-f005:**
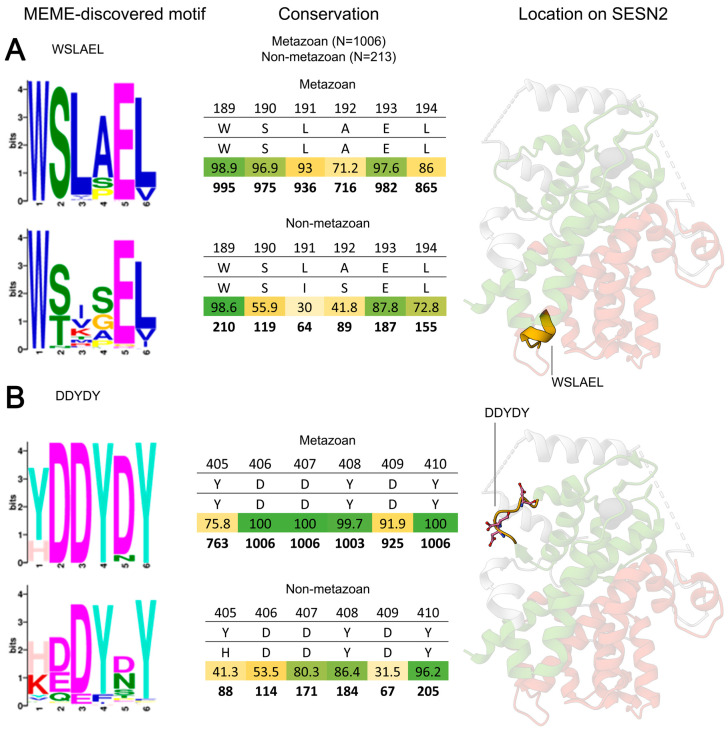
Conservation of Sestrin motifs implicated in GATOR2 binding. Motif conservation analysis displayed as in Figure 3. The motifs that unify the two domains for a single function. (**A**) WSLAEL is the site of GATOR2 binding on the SESN-A domain. (**B**) DDYDY is the key binding site of GATOR2 on the SESN-C domain.

**Figure 6 cells-13-01587-f006:**
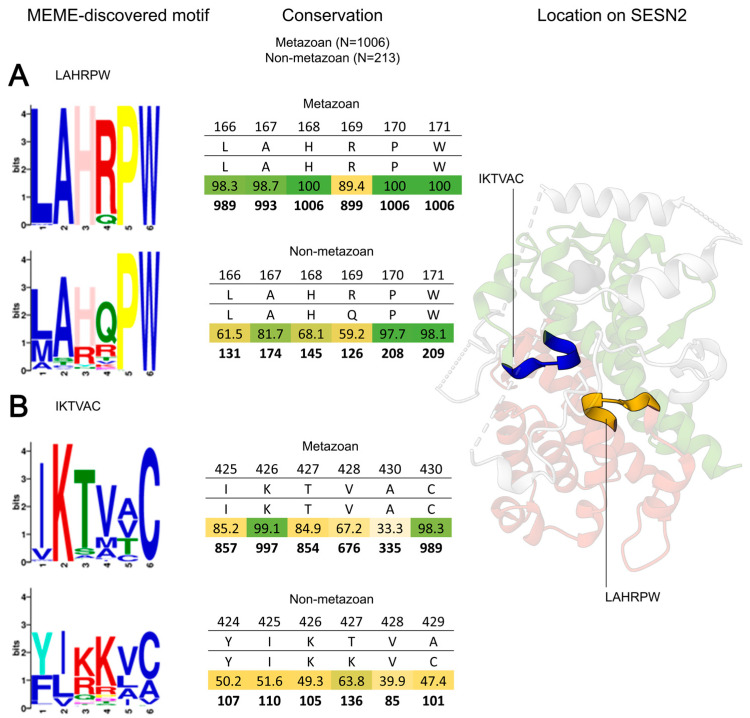
Conservation of the LAHRPW motif and its analogue IKTVAC. Motif conservation displayed as in Figure 3 and Figure 5. (**A**) Motif LAHRPW on the SESN-A domain is conserved but its function is unknown. (**B**) Motif IKTVAC is the analogous counterpart of LAHRPW on the SESN-C domain. Its position and structure are similar but its conservation is not as significant as that of LAHRPW.

**Figure 7 cells-13-01587-f007:**
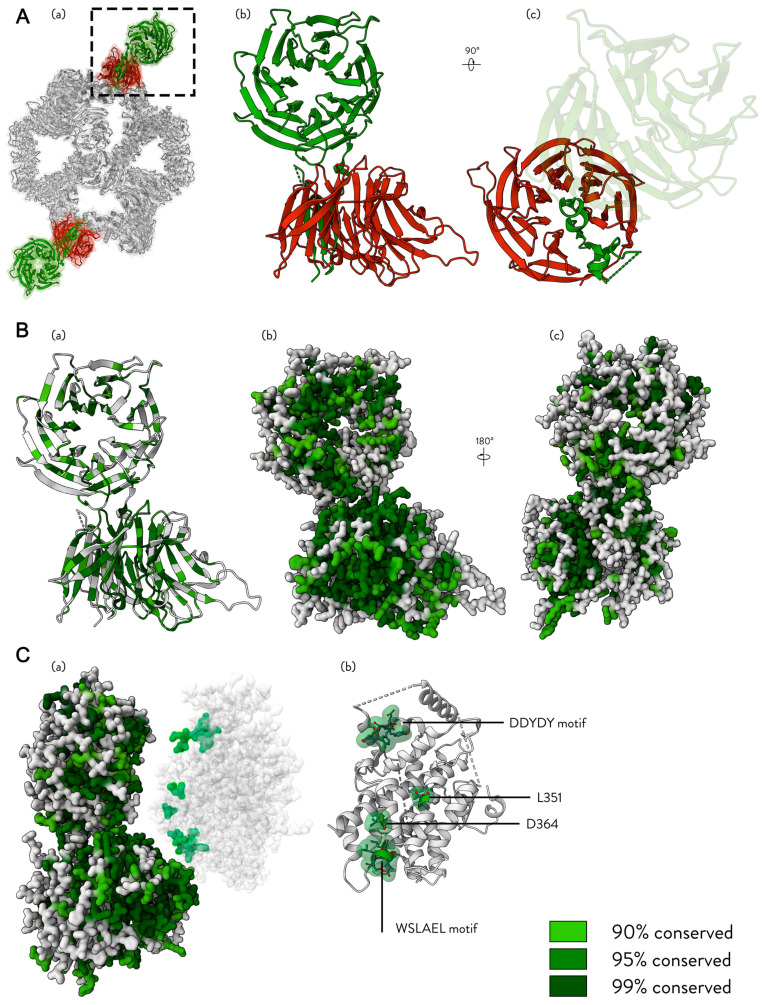
Sestrins bind somewhere along the WDR24-SEH1L β-propellers. The figure displays different representations of the WDR24-SEH1L β-propeller domains. (**A**) (**a**) The location of the WDR24-SEH1L β-propellers on the GATOR2 structure (PDB ID: 7UHY). (**b**) The cartoon representation of WDR24-SEH1L β-propellers. Green—WDR24; Red—SEH1L. (**c**) A top-down view of the blade donation from WDR24 to SEH1L, corresponding to (**b**) rotated by 90 degrees. The rest of the WDR24 structure is faded for visual clarity. (**B**) The results of our alignment analysis are visualised as shades of green on the WDR24-SEH1L arrangement. The legend for the shades of colour is displayed in the corner of the figure. (**a**) The cartoon representation of WDR24-SEH1L β-propellers and their conserved residues. (**b**) Surface representation of residues from the image in (**a**). (**c**) A view of the back of the arrangement, where (**b**) has been rotated along the y-axis by 180 degrees. (**C**) The hypothetical arrangement of the SESN2-GATOR2 binding. (**a**) A side view of the WDR24-SEH1L arrangement; the view above in (**B**) has been rotated by 90 degrees to the right along the y-axis. (**b**) The cartoon representation of the SESN2 face that binds GATOR2, with important sites annotated.

## Data Availability

The datasets generated during and/or analysed during the current study are available from the corresponding author on reasonable request.

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
