# Peer review of "Locked in Structure: Sestrin and GATOR—A Billion-Year Marriage"

_cells, 2024, doi:10.3390/cells13181587_

Round 1

Reviewer 1 Report

Comments and Suggestions for Authors

In the review article, Haidurov et al., has covered an extensive overview of the Sestrin proteins, through structural, evolutionary viewpoints. The authors also highlighted the Sestrin’s ability of sensing leucine and how it affects its interactions with GATOR2, which is crucial for regulating the mTORC1 signaling. However, this extensive review needs substantial improvement before further consideration.

1.        The review article has countless errors, indicating a significant lack of attention to the details before submitting the manuscript. Its’s the author’s responsibility to thoroughly proofread their document before submission rather than relying on reviewers to identify basic errors. The authors are strongly encouraged to do a comprehensive review of their article for any small errors. This is the minimum expectation before submitting any manuscript and should not be overlooked.

Here are a few examples,

·      From Figure 2; 2C legend is missing in the figure.

·      Figure 3; In the figure its only C, however it should A and B

·      “A further 10% present a glutamic acid here, showing that approximately 88% of non-metazoan Sestrins have an electronegative residue in this position (Figure 2C).” The reference Figure 2C is incorrect.

·      “The second best conserved motif after the DDYDY motif is LAHRPW and there is no evidence that it serves a unique function (Figure 2D).” The reference Figure 2D is incorrect.

·      Again in Figure 5, legends are wrong. Why there are two separate panels for Metazoan and non Metazon in the “LAHRPW” section?

·      “Mapping all residues with evidence of GATOR2-SESN2 interaction, Figure 4C (a) displays a hypothetical orientation for this interaction, while Figure 4C (b) shows that these residues are located along the same face of SESN2.” Figure citation is wrong.

2.        The review covers a comprehensive overview of Sestrins, however its excessive length and monotonous writing style are significant drawbacks. With a length of 19 pages with very repetitive theme and multiple similar analogy figures (Fig 2, 3, 5), it risks losing the reader’s attention. Some of sections, such as detailed structural analysis could be condensed.

3.        In the introduction area, the author should include a schematic illustrating the general biological implications of Sustrins, emphasizing  how leucine binding influences its interactions with other partners and eventually function.  This schematic should be designed in a way that it should give the readers a clear roadmap about what to expect from the review.

4.        The abstract is long and detailed. It can be compacted. Clearly state the purpose of the review.

5.        Add a discussion section highlighting major unanswered questions and future direction in the field.

6.        There should be a concluding section summering the few take home points.

7.        Although it is briefly discussed, please make a section on how different intracellular conditions changes intracellular Leucine concentration before discussing the leucine binding motifs.

8.        There are places, where a Figure reference would be helpful. One such example is,

·      Sestrins can be described as multifaceted proteins. They are described to hold three domains: SESN-A (or N-terminal domain), SESN-B (linker domain) and SESN-C (or C-terminal domain). Refer to Fig 1 here.

9.        The title is a bit misleading as the review primarily focused on structural analysis and evolutionarily perspective, with less emphasis on the Sustrins and GATOR biological aspects. Consider reviewing the title and content to reflect these points.

10.  Overall, the English needs to be improved on this article. There are many places the writing switched between past and present tense. Also maintain consistent formatting.

Comments on the Quality of English Language

There are places where it needs improvement in english. 

Author Response

  1. The review article has countless errors, indicating a significant lack of attention to the details before submitting the manuscript. Its’s the author’s responsibility to thoroughly proofread their document before submission rather than relying on reviewers to identify basic errors. The authors are strongly encouraged to do a comprehensive review of their article for any small errors. This is the minimum expectation before submitting any manuscript and should not be overlooked.

Here are a few examples,

  • From Figure 2; 2C legend is missing in the figure.
  • Figure 3; In the figure its only C, however it should A and B
  • “A further 10% present a glutamic acid here, showing that approximately 88% of non-metazoan Sestrins have an electronegative residue in this position (Figure 2C).” The reference Figure 2C is incorrect.
  • “The second best conserved motif after the DDYDY motif is LAHRPW and there is no evidence that it serves a unique function (Figure 2D).” The reference Figure 2D is incorrect.
  • Again in Figure 5, legends are wrong. Why there are two separate panels for Metazoan and non Metazon in the “LAHRPW” section?
  • “Mapping all residues with evidence of GATOR2-SESN2 interaction, Figure 4C (a) displays a hypothetical orientation for this interaction, while Figure 4C (b) shows that these residues are located along the same face of SESN2.” Figure citation is wrong.

We apologise for the quality of the first manuscript and we took care to correct all errors and improve the quality of language, the legends and figures.

  1. The review covers a comprehensive overview of Sestrins, however its excessive length and monotonous writing style are significant drawbacks. With a length of 19 pages with very repetitive theme and multiple similar analogy figures (Fig 2, 3, 5), it risks losing the reader’s attention. Some of sections, such as detailed structural analysis could be condensed.

We thank the reviewer for this insight. In line with this recommendation, we deleted the section Other Conserved Motifs with No Known Function and condensed it to a section about the extremely conserved LAHRPW motif. We attempted to leave only the most interesting information and believe that the conservation analysis section brings new insights into Sestrins. With reworked figures, and attached Excel files as Supplementary Files, it is now easier to follow.

  1. In the introduction area, the author should include a schematic illustrating the general biological implications of Sustrins, emphasizing  how leucine binding influences its interactions with other partners and eventually function.  This schematic should be designed in a way that it should give the readers a clear roadmap about what to expect from the review.

We thank the reviewer for this great suggestion – It is a great way to improve the quality of the review. A new picture, Figure 1, was added to address this recommendation. We hope you find the Figure 1 informative!

  1. The abstract is long and detailed. It can be compacted. Clearly state the purpose of the review.

In line with this recommendation, we shortened the abstract by approx. 70 words, making it four lines shorter and more concise. We also stated the purpose of this review in the abstract.

  1. Add a discussion section highlighting major unanswered questions and future direction in the field.

To address this, the last section Sestrins have co-evolved with GATOR2 has been restructured, repurposed, and supplemented. The new Discussion is a concise section that highlights and discusses the findings of the review. A piece with the most burning questions regarding this branch of research is also included.

  1. There should be a concluding section summering the few take home points.

We thank the reviewer for this recommendation. A short conclusion was added to summarise the key points of the review.

  1. Although it is briefly discussed, please make a section on how different intracellular conditions changes intracellular Leucine concentration before discussing the leucine binding motifs.

In line with this recommendation, we have added a new section before discussing the TYNT and HSEKVH motifs in the conservation section. This section covers how leucine concentrations fluctuate under different conditions, such as nutrient-rich environments and starvation. We included a relevant study, showing how leucine levels drop during starvation, leading to Sestrin-mediated inhibition of mTORC1.

  1. There are places, where a Figure reference would be helpful. One such example is,

  • Sestrins can be described as multifaceted proteins. They are described to hold three domains: SESN-A (or N-terminal domain), SESN-B (linker domain) and SESN-C (or C-terminal domain). Refer to Fig 1 here.

              We thank the reviewer for this recommendation and we added the recommended Figure     reference, as well in other places where such addition seemed appropriate.

  1. The title is a bit misleading as the review primarily focused on structural analysis and evolutionarily perspective, with less emphasis on the Sustrins and GATOR biological aspects. Consider reviewing the title and content to reflect these points.

We appreciate this comment and we have reconsidered the title. To better address that our review is based on structural analysis, we retitled the review
Locked in Structure: Sestrin and GATOR — A Billion-Year Marriage

Additionally, the revised discussion and conclusion underscores the idea that the most conserved structural components of Sestrins are those involved with GATOR, reflecting the updated title.

  1. Overall, the English needs to be improved on this article. There are many places the writing switched between past and present tense. Also maintain consistent formatting.

We took a great effort to rewrite the manuscript’s poor sections, and we hope that the new manuscript has fully addressed this concern. We thank you for your help and advice!

Reviewer 2 Report

Comments and Suggestions for Authors

The manuscript entitled: "Sestrin and GATOR: a billion-year marriage" by Alexander Haidurov and Andrei Budanov is a well-design and well-written review study regarding the nature of sestrins family of proteins and its interaction with GATOR complex regulating mTORC1-related processes.

I found the study either very interesting or well-planned. The study contains great figures which help understanding of proteomic data. It is also worth noting that the study emphasizes the role of in silico approaches to determine the meaning of complicated biological mechanisms and systems.

I have no major concerns regarding the study. I would only suggest to correct some editorial issues which I noted: lines 29 and 30 - there is no need for "and" and "in" being italicized within the brackets. Also please check the entire manuscript for errors in species binominal nomenclature - only the generic name should start with capitalised letter, whereas I noted e.g. Naeseria Gruberi.

Great work. It was a pleasure the read such a complite manuscript.

Author Response

I have no major concerns regarding the study. I would only suggest to correct some editorial issues which I noted: lines 29 and 30 - there is no need for "and" and "in" being italicized within the brackets. Also please check the entire manuscript for errors in species binominal nomenclature - only the generic name should start with capitalised letter, whereas I noted e.g. Naeseria Gruberi

We sincerely thank you for your kind words! In line with your recommendation, we attempted to eliminate all editorial issues, including the one that you brought up. Thank you for your help and we hope that you find the new manuscript even more interesting than our last!

Round 2

Reviewer 1 Report

Comments and Suggestions for Authors

The authors have adequately addressed the issues and improved the manuscript. It can be published now. 

Author Response

Thank you for your positive feedback and for recommending the manuscript for publication. We also appreciate your valuable suggestions and comments, which helped improve the manuscript.